# Output Characteristics of Carbon Nanotube Thermoelectric Generator with Slitted Kirigami Structure

**DOI:** 10.3390/ma18030656

**Published:** 2025-02-02

**Authors:** Shingo Terashima, Yuki Iwasa, Naoki Tanaka, Tsuyohiko Fujigaya, Eiji Iwase

**Affiliations:** 1Department of Applied Mechanics and Aerospace Engineering, Waseda University, 3-4-1 Okubo, Shinjuku, Tokyo 169-8555, Japan; iwasa@iwaselab.amech.waseda.ac.jp; 2Department of Applied Chemistry, Kyushu University, 744 Motooka, Nishi-ku, Fukuoka 819-0395, Japan; tanaka.naoki.468@m.kyushu-u.ac.jp (N.T.); fujigaya.tsuyohiko.948@m.kyushu-u.ac.jp (T.F.); 3Kagami Memorial Research Institute for Materials Science and Technology, Waseda University, 2-8-26 Nishiwaseda, Shinjuku, Tokyo 169-0051, Japan

**Keywords:** thermoelectric generator, single-walled carbon nanotube, energy harvesting, kirigami

## Abstract

The objective of our research is to improve the power generation of a thermoelectric generator (TEG) using a single-walled carbon nanotube (SWCNT) sheet by applying the out-of-plane deformation of a slitted kirigami structure. In order to obtain a large amount of power from a TEG using a thin-film thermoelectric (TE) element such as a SWCNT sheet, it is necessary to generate a large temperature difference in the in-plane direction of the thin-film TE element. However, it is difficult to realize a large temperature difference when the thin-film TE element is in contact with a heat source due to the need for a layer with high heat insulation. In this research, we proposed and fabricated a TEG with the out-of-plane deformation of a kirigami structure with slits using a p-n patterned SWCNT sheet as the thin-film TE material and evaluated the open circuit voltage with respect to the out-of-plane deformation and the number of TE elements. As a result, the output performance of SWCNT TEG was clarified when the out-of-plane deformation and the number of TE element pairs were varied.

## 1. Introduction

The objective of this research is to realize a large temperature difference within a single-walled carbon nanotube (SWCNT)-based thermoelectric (TE) element by the out-of-plane deformation of SWCNT sheets with slits, and to improve the output power of a thermoelectric generator (TEG). Thermoelectric conversion is an effective technology for sustainable energy utilization, which enables the production of electricity from waste heat. In conventional TEGs, inorganic materials such as Bi_2_Te_3_ [1,2,3], GeTe [4,5,6], PbSe [7,8], and others are well-known for their excellent conversion efficiency, which is expressed by the figure of merit (*ZT*).

However, they are rigid and cannot be bent, so they cannot be installed on curved heat sources [1,2,3,4,5,6,7,8,9,10,11,12,13,14,15,16,17,18,19,20,21,22,23]. To address this problem, designing thin TEGs has been proposed [16,17,18,19,20,21,22,23]. Organic and inorganic thin-film TE materials such as polymers like poly(3,-4-ethylenedioxythiophene):poly (styrene sulfonate) (PEDOT/PSS) and a SWCNT are being actively developed that do not use rare metals and can reduce manufacturing costs [24,25,26,27]. Among organic TE materials, SWCNTs are attracting attention because of their relatively high Seebeck coefficient and electrical conductivity. However, practical application has not been realized due to the lack of proposed device structures capable of achieving significant temperature differences within SWCNTs. As a TE element is thin, it is difficult to cause a temperature difference in the thickness direction, so realizing a temperature difference in the in-plane direction is necessary. In other words, the direction of heat flow needs to be changed after attaching the thin-film TE element to the heat source surface. Several structures that can change the direction of heat flow have been proposed so far [12,13,14,28,29]. However, considering changing the direction of heat flow makes the device structure complicated. Specifically, a high-temperature side of the TEG should be connected to the heat source, while a low-temperature side should be connected through heat insulation. On the low-temperature side, even though it is a material with high heat insulation, cooling the low-temperature side is difficult because the low-temperature side is closely connected to the heat source. Therefore, in this research, we considered increasing the temperature difference in the in-plane direction of the SWCNT TE element by promoting heat dissipation on the low-temperature side by out-of-plane deforming by a kirigami structure as shown in Figure 1. The proposed TEG has the kirigami structure with slits, so this TEG is called a slitted kirigami TEG. In this study, we have evaluated the output performance of the slitted kirigami TEG when the amount of out-of-plane deformation in the slitted kirigami TEG and the number of divisions in the slitted kirigami TEG were changed.

## 2. Materials and Methods

### 2.1. Material

Figure 1 shows a schematic diagram of the slitted kirigami TEG used in the experiment. As a TE element, a SWCNT sheet with half the p-type region and half the n-type region was used as shown in Figure 1. Then, we connect to copper (Cu) electrodes with a conductive adhesive as shown in Figure 1.

The fabrication process of the SWCNT sheet is as follows. SWCNTs with a diameter of 1.5 ± 0.5 nm (Meijo-e-DIPS, Meijo Nano Carbon, Nagoya, Japan, 80 mg) were dispersed in N-methylpyrrolidone (NMP) (1000 mL) using High Shear Mixers (SILVERSON, Bucks, UK) at 8000 rpm for 10 min × 3 sets. SWCNT films were obtained by filtering the dispersion through a polytetrafluoroethylene membrane (diameter: 90 mm, pore size: 3.0 μm). The obtained films on the membrane were peeled off and immersed in methanol to remove residual NMP for 12 h, and then dried at 100 °C for 8 h under a vacuum (1 Torr). The thickness of the films was around 47 ± 3 μm. After cutting the SWCNT films into square shapes (50 mm × 50 mm), half of the area was n-doped by spray-coating with a 5.0 wt% ethanol solution of polyethylene-imine (PEI), while the other half was covered with a glass slide as a mask. This process created a pair of p-type and n-type regions, as the masked area retained p-type characteristics due to the auto-oxidation of SWCNTs by oxygen. The Seebeck coefficient was recorded under air at a temperature of 30 °C using a ZEM-3 measurement system (ADVANCE RIKO, Yokohama, Japan) to evaluate the carrier characteristics of the p-type and n-type regions. Positive and negative Seebeck coefficients indicate the p-type or n-type properties of the conductor, respectively. The Seebeck coefficients of 69 μV/K for the p-type region and −34 μV/K for the n-type region confirmed the n-doping of the area by PEI.

In case a SWCNT sheet is used for its TE elements, the contact resistance between the SWCNT sheet as the TE elements and the lead electrodes is important. For inorganic TE materials such a bismuth telluride, some techniques for connecting them to electrodes are suggested. However, for organic TE materials, such techniques have not been developed, and it is still unclear how to minimize contact resistance. Therefore, we investigated the volume resistance of each component material and the contact resistance between the components of a device in which SWCNTs and metal electrodes are connected with electrical connectors. A device for measuring resistance was fabricated using copper electrodes as metal electrodes and a conductive adhesive as an electrical connection material as shown in Figure 2. To fabricate the device, a SWCNT sheet with a thickness of 53.4 µm was used. As the electrode, a film consisting of an 8 µm thick copper layer on a 25 µm thick polyimide substrate (Toray Advanced Materials Korea Inc., Tokyo, Japan) was selected. For electrical connections, a conductive adhesive (Fujikura kasei Co., Ltd., Tokyo, Japan) was applied. A slide glass with a thickness of 1.5 mm was applied as the substrate, and polyimide tape was used to mask the area around the connection areas. The fabrication process began by attaching the polyimide-copper film to the entire upper surface of the glass substrate with double-sided tape, positioning the copper layer facing up. Next, the attached polyimide-copper film was cut into the shape of the electrode using a UV laser processing machine (OPI Co., Ltd., Hidaka, Japan), and the unnecessary parts were removed with tweezers. At this time, the double-sided tape was also peeled off from the glass substrate. A SWCNT sheet cut to 2 mm × 18 mm using a UV laser processing machine was placed between the copper electrodes, and the SWCNT sheet and copper electrodes were masked with polyimide tape. At this time, the mask was placed so that the contact area between the copper electrode and the conductive adhesive was 2 mm wide and 1 mm long (*L*_Cu-ad_), and the contact area between the SWCNT sheet and the conductive adhesive was 2 mm wide and 4 mm long (*L*_CNT-ad_). After that, the conductive adhesive was placed and cured by heating for 1 h at 170 °C in a constant temperature incubator (Yamato Scientific Co., Ltd., Tokyo, Japan). Four similar devices were fabricated. The electrical resistance was measured 100 times using a source meter (Keithey Instruments, Tokyo, Japan) when a current of 0.1 A was passed through the device, and the average value was calculated. Using the four-terminal method, the electrical resistance of the entire device (electrical resistance between the copper electrodes) *R*_1_ and the electrical resistance *R*_2_ between the two conductive adhesives were measured. The average measured values of *R*_1_ and *R*_2_ for the four devices were *R*_1_ = 1.48 Ω and *R*_2_ = 1.45 Ω. Here, the volume resistance of one copper electrode is defined as *R*_Cu_, the volume resistance of one conductive adhesive as *R*_ad_, the volume resistance of the SWCNT sheet as *R*_CNT_, the contact resistance of one connection between the copper electrode and the conductive adhesive is *R*_Cu-ad_, and the contact resistance of one connection between the SWCNT sheet and the conductive adhesive is *R*_CNT-ad_, and *R*_1_ and *R*_2_ can be expressed as *R*_1_ = 2*R*_Cu_ + 2*R*_Cu-ad_ +2*R*_ad_ + 2*R*_CNT-ad_ + *R*_CNT_, and *R*_2_ = 2*R*_ad_ + 2*R*_CNT-ad_ + *R*_CNT_. The volume resistivity of the SWCNT sheet is 1.6 × 10^−7^ Ω m (the electrical conductivity is 6.25 × 10^6^ S/m). The volume resistivities of the copper electrode and conductive adhesive are 1.7 × 10^−8^ Ω m and 8.0 × 10^−7^ Ω m, respectively. The length of the copper electrode is 16 mm, and the thickness of the conductive adhesive is 1 mm. The calculated electrical resistances are *R*_Cu_ = 1.3 × 10^−2^ Ω, *R*_ad_ = 2.0 × 10^−3^ Ω, and *R*_CNT_ = 2.7 × 10^−2^ Ω. Since *R*_Cu_ and *R*_CNT_ on the order of 10^−2^ Ω and *R*_ad_ on the order of 10^−3^ Ω are sufficiently small compared to *R*_1_ and *R*_2_ on the order of 10^0^ Ω, *R*_1_ and *R*_2_ can be expressed as *R*_1_ = 2*R*_Cu-ad_ + 2*R*_CNT-ad_ and *R*_2_ = 2*R*_CNT-ad_. The measurement results and *R*_1_ and *R*_2_ above show that the contact resistance *R*_CNT-ad_ between the conductive adhesive and the SWCNT sheet accounts for approximately 95% of the electrical resistance of the entire device, and that in order to reduce the electrical resistance of the entire device, it is effective to use a connection method that reduces the contact resistance *R*_CNT-ad_ between the conductive adhesive and the SWCNT sheet.

Next, to evaluate the electrical resistance depending on the type of electrical connecting material, devices were fabricated using different connecting materials and their electrical resistance was measured. A total of four types of devices were compared: devices using three types of electrical connecting materials—solder (TAIYO ELECTRIC IND. Co., Ltd., Fukuyama, Japan), liquid metal (Galinstan), and conductive adhesive—and a device without any connecting materials. Galinstan is a liquid metal alloy at room temperature, primarily composed of gallium (Ga), indium (In), and tin (Sn). In these devices, the materials used are the same as those used in the previously described devices, except for the electrical connections. Here, a SWCNT sheet cut to 2 mm × 10 mm was used, and the distance between the copper electrodes was set to 6 mm, with part of the SWCNT sheet overlapping with the copper electrodes. The fabrication method of the device is the same as that of the previously described device. For devices without electrical connections, the SWCNT sheet was contacted with the copper electrode using water, and then dried and connected at room temperature. The current was measured when the voltage was swept from −1 V to 1 V in 100 steps using a source meter, and the electrical resistance was obtained from the slope of the linear approximation. The measurements were performed using the four-terminal method. As the measurement results, when using solder, liquid metal, and conductive adhesive, the contact resistance was 2.6 Ω, 1.3 Ω, and 1.1 Ω, respectively. And the contact resistance without an electrical connecting material was the highest at 6.7 Ω. In the case of using liquid metal or conductive adhesive, the contact resistance is low. After hardening, conductive adhesive is solid at room temperature, whereas liquid metal is liquid. Therefore, taking into consideration the strength of the connected part after hardening, we decided that conductive adhesive is suitable for TEG. Therefore, we used the conductive adhesive for the conducting material between the SWCNT sheet and the Cu electrodes. To cure the conductive adhesive, we heated it at 170 °C for 1 h in an oven. After that, epoxy resin (Konishi Co., Ltd., Osaka, Japan) was applied to increase the strength of the connection between the lead wires and the conductive adhesive, and left to cure at room temperature for one day. In case the contact area between the SWCNT sheet and the conductive adhesive is 2 mm × 2 mm, the contact resistance was 1.1 Ω.

In order to evaluate the contact resistance depending on the dimensions of the connection area between the SWCNT sheet and the electrical connection material, devices with different lengths and widths of the connection were fabricated and their contact resistance was measured. First, the contact resistance when the length of the contact part between the SWCNT sheet and the conductive adhesive is changed is discussed. The same device as above was used to measure the contact resistance. The material and fabrication method of the device were the same, and the width of one point of the contact part between the SWCNT sheet and the conductive adhesive was set to 2 mm. Four types of devices were fabricated in four lengths: *L*_CNT-ad_ of 1, 2, 3, and 4 mm. The contact resistance between the SWCNT sheet and the conductive adhesive was calculated by measuring the electrical resistance *R*_2_ between the conductive adhesives and subtracting the volume resistance of the two conductive adhesives 2*R*_ad_ and the volume resistance of the SWCNT sheet *R*_CNT_. When the length of the connection part was changed, the contact resistance decreased as the length of the connection increased, reaching a maximum of 1.12 Ω at 1 mm and a minimum of 0.72 Ω at 4 mm. Therefore, to reduce the contact resistance, the length of the connection part should be increased. The contact resistance, however, became a larger resistance than 1/*n* in the case that the length of the connection was increased by *n* times. That is, when the contact resistance per length of connection part was calculated, it was found that the longer the contact length leads to the larger contact resistance per length of the connection part. The length of the connection part should be shortened to reduce the contact resistance per unit area.

Next, the contact resistance when the width of the contact area is changed is discussed. The device used to measure the contact resistance is the same as described above. The length of the connection part between the SWCNT sheet and the conductive adhesive was fixed at 2 mm, and devices with different widths of 2 and 10 mm were fabricated. Using the same measurement method as above, the electrical resistance *R*_2_ between the two conductive adhesives was measured, and the contact resistance between the SWCNT sheet and the conductive adhesive was calculated by subtracting the volume resistance of the two conductive adhesives 2*R*_ad_ and the volume resistance of the SWCNT sheet *R*_CNT_. As a result, it was found that the contact resistance was 1.08 Ω at 2 mm of the contact width and 0.36 Ω at 10 mm of the contact width, and that the wider the contact width, the smaller the contact resistance. The same as in the case where the length of the connection was changed, the contact resistance became a larger resistance than 1/*n* even in the case that the width of the connection was increased by *n* times. Therefore, the width of the connection should be narrowed to reduce the contact resistance per contact area.

### 2.2. Methods

As shown in Figure 1, by passing a silicone rubber string through the slits of the SWCNT sheet and deforming in the out-of-plane direction, it is possible to give an appropriate directional temperature gradient to each p-type and n-type region. In the slitted kirigami TEG, silicone rubber string plays the role of both out-of-plane deformation and heat insulation. The slitted kirigami TEG uses the temperature gradient that occurs in the height direction. Specifically, the lower part of the SWCNT in contact with the heat source is the high-temperature side, and the upper part away from the heat source is the low-temperature side. The insulating material between the lower and upper part should have low adiabatic thermal conductivity. Also, considering the output performance test with respect to the amount of out-of-plane deformation, a shape that can fix the amount of out-of-plane deformation is desirable, so we chose a cylindrical shape. For example, wooden rods, plastic rods, and rubber strings satisfy these criteria. We selected silicone rubber string, which is easy to obtain in various diameters and is easy to cut. As a result, a temperature difference occurs within the SWCNT TE element, and output power is obtained. The fabrication process and the photograph of the fabricated TEG are shown in Figure 3. As shown Figure 3b, the SWCNT sheet used in this experiment has a thickness of 47 ± 3 µm and patterned regions of half p-type and half n-type. We cut a symmetrical slit pattern into the SWCNT sheet using a UV laser (OPI Co., Ltd., Hidaka, Japan). To maintain good thermal contact on the heating side of the SWCNT TE element during the experiment, the slitted kirigami TEG was fixed on a 500 μm thick aluminum plate using a thermally conductive adhesive (Cemedine Co. Ltd., Tokyo, Japan). This aluminum plate was placed on the heater using a thermally conductive grease (Shin-Etsu Silicone, Tokyo, Japan). After these processes, two lead wires are mounted on both ends of the slitted kirigami TEG with the conductive adhesive (Fujikura kasei Co., Ltd., Tokyo, Japan). Finally, a cylindrical silicone rubber string is inserted to deform the SWCNT sheet out-of-plane and insulate the heat from the heater. The current-voltage characteristics were measured using the setup shown in Figure 3d.

In order to evaluate the power generation performance of the slitted kirigami TEG depending on the height of the out-of-plane deformation, slitted kirigami TEGs with different heights were fabricated. As shown in Figure 4a, the SWCNT sheet is cut into 6 mm × 24 mm (p-type region: 6 mm × 12 mm, n-type region: 6 mm × 12 mm) rectangles with alternating 22 mm long slits from the left and right sides to form a 2 mm wide path. By changing the diameter of the silicone rubber string, the height of the slitted structure can be changed as shown in Figure 4b. The silicone rubber diameters were changed to 2, 4, 6, 8, 10, and 12 mm.

Next, in order to evaluate the power generation performance of the slitted kirigami TEG depending on the number of pairs of SWCNT TE elements using the same area of the SWCNT sheet, slitted kirigami TEGs of variable beam widths were fabricated. As shown Figure 5a, the SWCNT sheet is cut into 24 mm × 36 mm (p-type region: 12 mm × 36 mm, n-type region: 12 mm × 36 mm) rectangles with alternating 22 mm long slits from the left and right sides to form a 2 mm wide path. The fabricated slitted kirigami TEGs for each number of pairs are shown in Figure 5b.

## 3. Results

### 3.1. Effect of Out-of-Plane Deformation

To evaluate the power generation performance of the slitted kirigami TEG depending on the height of the out-of-plane deformation, the current-voltage characteristics were measured when the height of the slitted structure was varied. The silicone rubber diameters were changed to 2, 4, 6, 8, 10, and 12 mm, and the current-voltage characteristics were measured. Here, the change in the diameter of the silicone rubber string corresponds to the change in the amount of out-of-plane deformation. As shown in Figure 6a, the electrical resistance was constant at 8.0–8.5 Ω regardless of the height of the out-of-plane deformation. The heating temperatures were 50, 70, and 90 °C, and the room temperature at the time of measurement was 25 °C. In this experiment, the slitted kirigami TEG was not forcedly cooled. In other words, the cold side of the slitted kigami TEG is exposed to room temperature (25 °C), and a temperature difference occurs within the TE element due to heat transfer to the outside air. Figure 6b,c show the measured open circuit voltage and maximum output power at different heights of the slitted kirigami structure. The open circuit voltage (Figure 6b) tended to increase with respect to the height of the slitted kirigami structure at all heating temperatures, reaching a maximum of 3.99 mV at 90 °C for a height of 12 mm. The out-of-plane deformation increases the temperature difference inside the SWCNT sheet. The higher the out-of-plane deformation, the more heat is dissipated by heat transfer to the surrounding air, and the larger the temperature difference is in the SWCNT sheet. As shown in Figure 6c, the maximum output power increased with respect to the height of the slitted kirigami structure at all heating temperatures, reaching a maximum of 446 nW at 90 °C of heating at a height of 12 mm. This is because increasing the height of the slitted kirigami structure increased the open circuit voltage while keeping the electrical resistance constant. These results indicated that increasing the diameter of the silicon rubber string, which is a heat insulator, and increasing the amount of out-of-plane deformation is effective in improving output performance.

### 3.2. Effect of the Number of Divisions in the Slitted Kirigami TEG

We evaluated the relationship between the number of divisions in the slitted kirigami TEG and the power generation performance under the condition that the area of the SWCNT sheet was the same. In a simple model, if the area of the SWCNT sheet is the same, the maximum output power should be the same. The reasons are as follows. We consider the case that the SWCNT sheet is divided into *n* parts. The resistance of the SWCNT sheet *R*_TEG_ should become *n*^2^ times larger because the width is 1/*n* times narrower and the length is *n* times longer. Under the same temperature difference, the open-circuit voltage *V*_TEG_ should become *n* times larger because the number of TE pair is *n* times bigger. Because the maximum output power can be calculated by ${V_{TEG}}^{2}/4R_{TEG}$, the maximum output power should be constant regardless of *n*. In this experiment, the out-of-plane deformation height of the silicon rubber string is 4 mm and the area of the SWCNT sheet is constant at 24 mm × 36 mm. The current-voltage characteristics of the slitted kirigami TEG are measured for different numbers of divisions in the slitted kirigami TEG. The number of SWCNT TE beams, which is the same number of divisions, was set to 3, 5, 9, and 19. Since the area of the SWCNT sheet is the same, the larger the number of beams, the smaller the beam width; when the number of beams was 3, 5, 9, and 19, the beam width was 12.0, 7.2, 4.0, and 1.8 mm, respectively. The heating temperatures were 50, 70, and 90 °C, and the room temperature at the time of measurement was 25 °C. Figure 7a–c show the measured electrical resistance, open circuit voltage, and maximum output power for different numbers of divisions in the slitted kirigami TEG. The electrical resistance ranged from 2.8 to 42.2 Ω, with a maximum at the narrowest beam width as shown in Figure 7a. When approximating using only *n*^2^ terms, the approximation formula was *R*_TEG_ = 0.10*n*^2^, with a coefficient of determination *R*^2^ = 0.9785. On the other hand, when approximating using n^2^ term and n^1^ term, the approximation formula was *R*_TEG_ = 0.054*n*^2^ + 0.90*n*, with a coefficient of determination *R*^2^ = 0.9997. Comparing the *R*^2^ for each approximation formula, the resistance was not proportional to *n*^2^ for the number of divisions *n*, but rather fitted *R*_TEG_ = 0.054*n*^2^ + 0.90*n* well. The term of the second power of *n* should be coming from the resistance of the SWCNT sheet as we described above as a simple model. The term of the first power of *n* can be understood to come from the contact resistance between the conductive adhesive and the SWCNT sheet. In the case that the number of divisions becomes *n,* the contact area of the conductive adhesive becomes 1/*n* times smaller and so the contact resistance of the conductive should be proportional to *n*. As shown in Figure 7b, the open-circuit voltage tended to increase linearly with the number of beams the slitted kirigami TEG, and reached a maximum of 21.8 mV at 90 °C when the number of beams was 19 and the beam width was 1.8 mm. The open-circuit voltage per pair at 90 °C was almost constant at 2.4–2.8 mV regardless of the beam width. As shown in Figure 7c, the maximum output power was greater for a larger number of p/n pairs and narrower beam widths at all heating temperatures, reaching a maximum of 2.4 μW at 90 °C when the number of beams was 19 and the beam width was 1.8 mm. Fujigaya et al. have proposed a structure that can create a temperature difference in a SWCNT, but the output is very small, at 5.7 nW at a temperature difference of 30 °C [28,29]. In contrast, the slitted kirigami TEG achieved a dramatic increase in output, at 450 nW, despite a temperature difference of 25 °C. These results show that increasing the voltage due to the large number of p/n pairs is the most effective way to improve output, if the contact resistance is not negligible.

## 4. Conclusions

In this study, we fabricated a slitted kirigami TEG using a SWCNT TE sheet, applied a kirigami structure, and investigated the output performance of it. The out-of-plane deformation of the slitted kirigami TEG was achieved by applying the slitted kirigami structure to the SWCNT TE sheet and then inserting a silicone rubber string into the slit in the kirigami structure. Before investigating the power generation performance of the slitted kirigami TEG, we clarified that the contact resistance at the interface between the SWCNT sheet and the conductive adhesive was large. Additionally, in order to reduce the contact resistance between the SWCNT sheet and the conductive connecting material, the contact resistance for four different connecting methods (solder, liquid metal, conductive adhesive, and without any connecting materials) was measured. As a result, we found that connecting the SWCNT sheet using a conductive adhesive was the best way to reduce the contact resistance. In the first experiment for the slitted kirigami TEG, the power generation performance of the slitted kirigami TEG was measured by varying the amount of out-of-plane deformation by changing the diameter of the silicone rubber string. In the results, the electrical resistance of the slitted kirigami TEG was constant regardless of the amount of out-of-plane deformation, and the temperature difference within the SWCNT TE element increased depending on the amount of out-of-plane deformation. Therefore, it was shown that designing a large amount of out-of-plane deformation leads to higher output power. In the next experiment, the power generation performance was measured by changing the beam width of the slitted kirigami TEG while keeping the installation area constant. As a result, a higher output was achieved when the number of divisions was large. We have shown that in order to achieve high output power, it is preferable to design as many divisions in the slitted kirigami TEG as possible within a limited installation area if the contact resistance is not negligible. From the above results, we succeeded in achieving high output by designing a device that increases the amount of the three-dimensional out-of-plane deformation of the slitted kirigami TEG and the number of pairs of the SWCNT TE elements. Unlike conventional TEGs, we hope to see the social implementation of harmless and low-cost TEGs. However, in the future, we will consider attaching them to human skin to monitor health conditions. We also believe that by attaching them to livestock such as cows, it will be possible to predict their estrus period.

## Figures and Tables

**Figure 1 materials-18-00656-f001:**
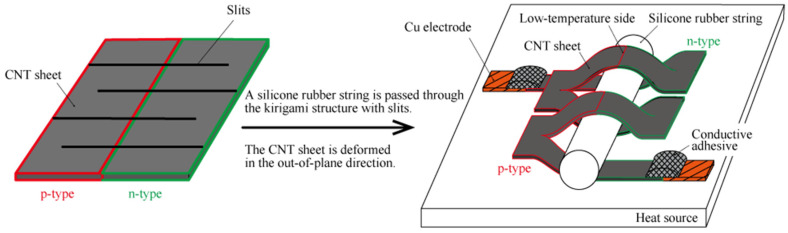
Basic structure of the slitted kirigami TEG. Half of the SWCNT sheet is a doped p-type region and the other half is a doped n-type region. A silicone rubber string is passed through the slits of the SWCNT sheet.

**Figure 2 materials-18-00656-f002:**
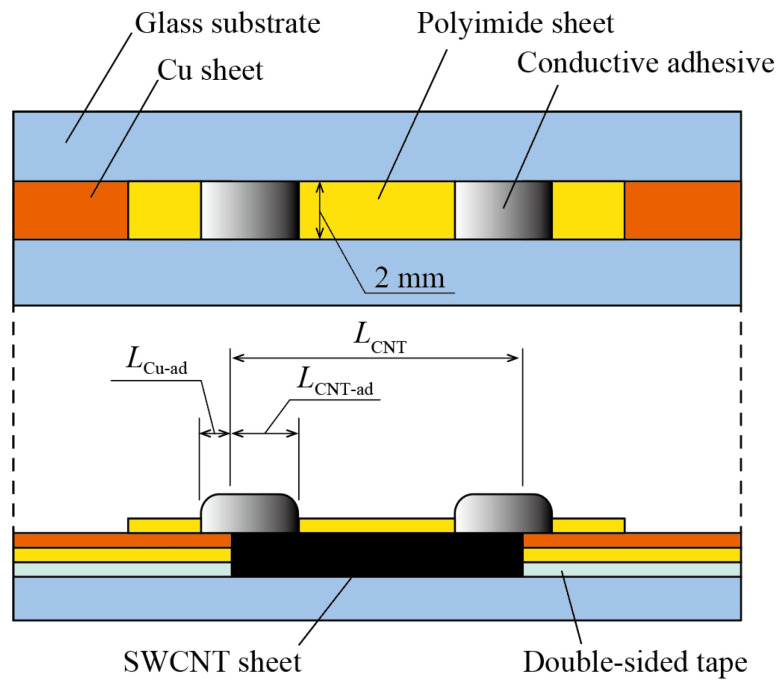
Top and side views of the contact resistance measurement device.

**Figure 3 materials-18-00656-f003:**
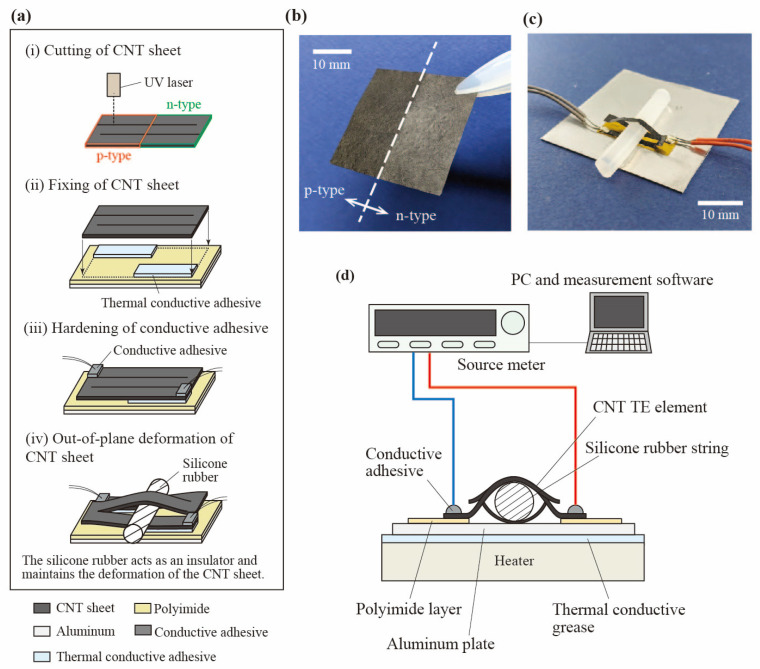
Fabrication process and fabricated TEG. (**a**) Fabrication process of the slitted kirigami TEG with a SWCNT sheet. (**b**) Photograph of the SWCNT sheet used in the slitted kirigami TEG. (**c**) Photograph of the fabricated slitted kirigami TEG. (**d**) Schematics of the setup to measure the current-voltage characteristics.

**Figure 4 materials-18-00656-f004:**
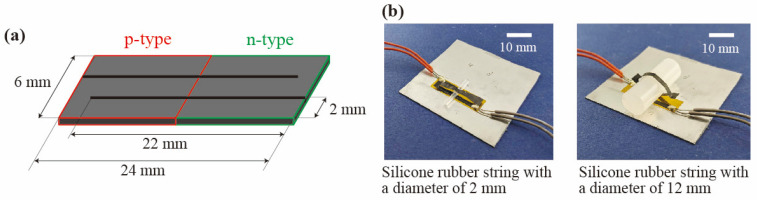
Evaluation method of the effect of out-of-plane deformation. (**a**) Dimensions of the SWCNT element used for the slitted kirigami TEG. (**b**) The slitted kirigami TEG used in the experiments.

**Figure 5 materials-18-00656-f005:**
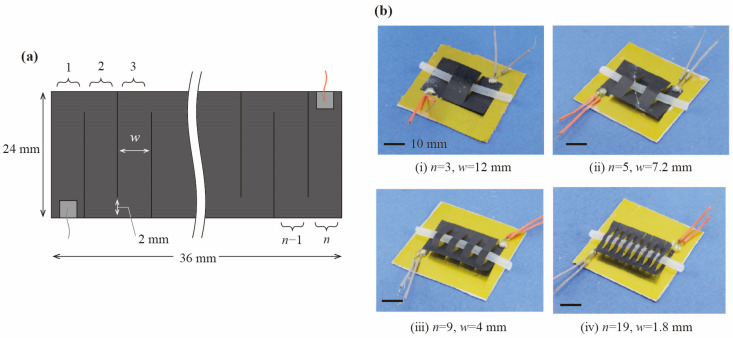
Evaluation method of the effect of the number of pairs of SWCNT TE elements. (**a**) Schematics of the slitted kirigami TEG. (**b**) Photograph of the fabricated slitted kirigami TEGs.

**Figure 6 materials-18-00656-f006:**
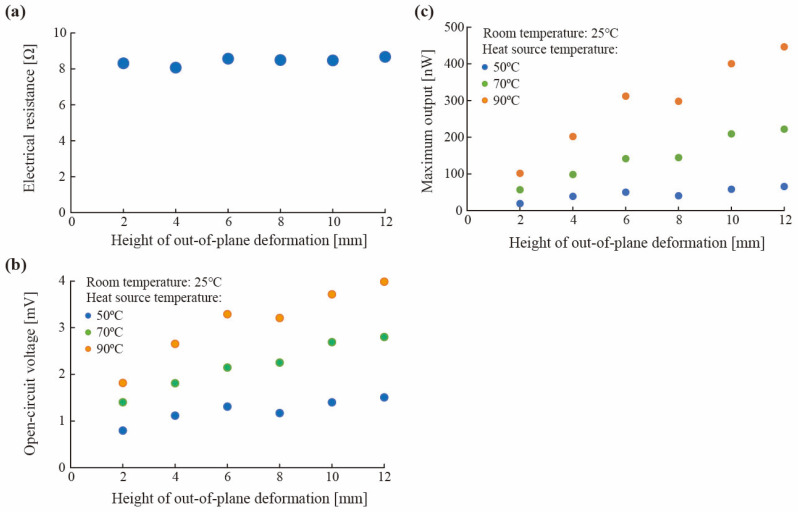
Evaluation results of the effect of out-of-plane deformation. (**a**) Electrical resistance with respect to the height of out-of-plane deformation. (**b**) Open-circuit voltage with respect to the height of out-of-plane deformation. (**c**) Maximum output with respect to the height of out-of-plane deformation.

**Figure 7 materials-18-00656-f007:**
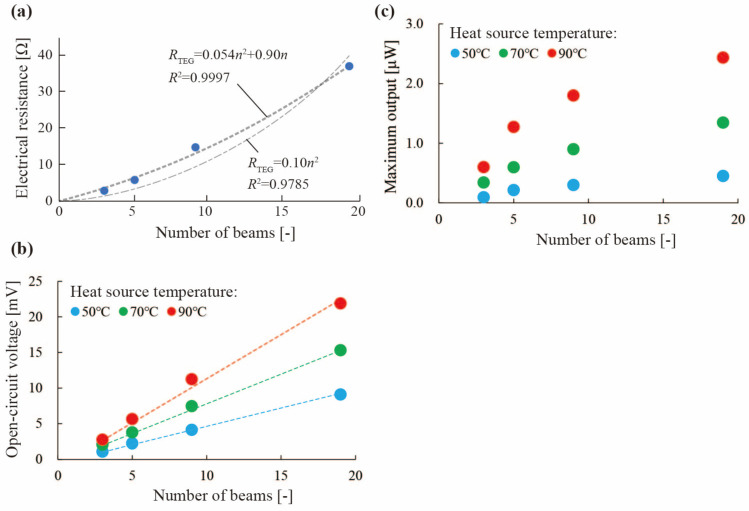
Evaluation results of the effect of the number of divisions in the slitted kirigami TEG. (**a**) Measured electrical resistance with respect to the number of SWCNT TE beams. (**b**) Measured open-circuit voltage with respect to the number of SWCNT TE beams. (**c**) Maximum output with respect to the number of SWCNT TE beams.

## Data Availability

The original contributions presented in this study are included in the article. Further inquiries can be directed to the corresponding author.

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
