# Peer review of "Output Characteristics of Carbon Nanotube Thermoelectric Generator with Slitted Kirigami Structure"

_materials, 2025, doi:10.3390/ma18030656_

Round 1
Reviewer 1 Report
Comments and Suggestions for Authors
This manuscript presents a novel approach to enhancing the power generation capabilities of a thermoelectric generator (TEG) by employing a single-walled carbon nanotube (SWCNT) sheet with a slitted kirigami structure. The topic is attractive and falls within the scope of Materials. The manuscript is well-organized, with a clear presentation of the research objectives, materials and methods, results, and conclusions. However, before publications some revisions are mandatory.
Ensure that the abstract succinctly captures the essence of the study, including the key findings and their implications. The detailed comments or suggestions are provided below.
1. Please provide more details on the fabrication process of the SWCNT sheet, including the parameters used during the high shear mixing and the vacuum drying process.
2. Please elaborate on the selection criteria for the silicone rubber string as the out-of-plane deformation material, including its thermal conductivity and mechanical properties.
3. It is necessary to include a control experiment or comparison with a conventional TEG to highlight the advantages of the proposed slitted kirigami TEG.
4. This research should consider using advanced data fitting techniques to analyze the relationship between the number of TE elements and the output performance.
5. Please ensure that all figures and schematics are clear and of high quality, with appropriate labeling and scaling.
6. The paper should discuss the potential impact of this research on the field of wearable electronics and energy harvesting, including possible commercial applications.
7. The paper should compare the performance of the slitted kirigami TEG with other state-of-the-art TEGs, highlighting the advantages and potential improvements.
8. Please ensure that the manuscript is free of grammatical errors and typos. Consider having a native English speaker or a professional editor review the manuscript for language clarity.
Comments on the Quality of English LanguageThe English could be improved to more clearly express the research.
Author Response
Thank you for your review.
Your comment was excellent and made it easier to explain our research findings.
We have summarized our responses in the attached PDF file. Please check it.

Reviewer 2 Report
Comments and Suggestions for Authors
Here, the authors employed out-of-plane deformation of a slitted kirigami structure to improve the power generation of a thermoelectric generator (TEG) based on single-walled carbon nanotube (SWCNT) sheets.
After carefully reading through the manuscript, I believe this study is interesting and worth publication after addressing some minor concerns as below:
1. Thermoelectric performance of the employed carbon nanotube sheets should be briefly discussed, such as electrical conductivity and Seebeck coefficient.
2. How are the p-type and n-type semiconducting behavior of carbon nanotube sheets realized? What dopants and techniques are used? These details should be briefly specified.
3. Typical inorganic thermoelectric materials, such as Bi2Te3 (10.1039/d3ee02370b) and GeTe (10.1021/jacs.3c12546), should also be briefly discussed and introduced for comparison with the organic counterparts.
4. Heat source temperature has been provided. However, key temperature-related parameter determining the power generating performance is the temperature difference. For this reason, the cold side temperature should also be provided.
Author Response
Thank you for your review.
Your comments were excellent and made it easier to explain our research findings.
We have summarized our responses in the attached PDF file. Please check it.

Reviewer 3 Report
Comments and Suggestions for Authors
Terashima et al. reported deformable thermoelectric devices with single-walled carbon nanotubes. Owing to the various form factor devices, the development of flexible or deformable energy harvesting devices is crucial. In this perspective, the authors approached the Kirigami structure with single-walled carbon nanotube based thermoelectric devices. Before publication, several issues should be addressed.
1. It seems that one of the developed device's advantages is that it is deformable under mechanical stress. Have the authors tried to characterize mechanical stability, such as through a cyclic bending test?
2. The device's performance is impressive. Have the authors considered the application, such as a wearable energy harvesting device?
3. In page 4, line 118, what is the exact meaning of volume resistivities? If the authors are talking about resistivity, the unit should be edited.
Author Response

(The authors gave the same response as above.)

Round 2
Reviewer 1 Report
Comments and Suggestions for Authors
All the issues were addressed. The paper can be published in its current form.